# Rho GTPases—Emerging Regulators of Glucose Homeostasis and Metabolic Health

**DOI:** 10.3390/cells8050434

**Published:** 2019-05-09

**Authors:** Lisbeth Liliendal Valbjørn Møller, Amira Klip, Lykke Sylow

**Affiliations:** 1Section of Molecular Physiology, Department of Nutrition, Exercise and Sports, Faculty of Science, University of Copenhagen, 2100 Copenhagen Oe, Denmark; Lmo@nexs.ku.dk; 2Cell Biology Program, The Hospital for Sick Children, Toronto, ON M5G 0A4, Canada; Amira@sickkids.ca

**Keywords:** Rho GTPases, metabolism, glucose homeostasis, GLUT4 translocation, skeletal muscle, pancreas, insulin, diabetes, ageing

## Abstract

Rho guanosine triphosphatases (GTPases) are key regulators in a number of cellular functions, including actin cytoskeleton remodeling and vesicle traffic. Traditionally, Rho GTPases are studied because of their function in cell migration and cancer, while their roles in metabolism are less documented. However, emerging evidence implicates Rho GTPases as regulators of processes of crucial importance for maintaining metabolic homeostasis. Thus, the time is now ripe for reviewing Rho GTPases in the context of metabolic health. Rho GTPase-mediated key processes include the release of insulin from pancreatic β cells, glucose uptake into skeletal muscle and adipose tissue, and muscle mass regulation. Through the current review, we cast light on the important roles of Rho GTPases in skeletal muscle, adipose tissue, and the pancreas and discuss the proposed mechanisms by which Rho GTPases act to regulate glucose metabolism in health and disease. We also describe challenges and goals for future research.

## 1. Introduction

The Rho family of small guanosine triphosphatases (GTPases) is a distinct branch within the superfamily of Ras-related small GTPases. Twenty mammalian genes encoding Rho GTPases have been identified of which Rac1, Cdc42, and RhoA are the prototypes and therefore the best characterized. Rho GTPases are crucial organizers of the actin cytoskeleton with essential functions in cell migration, cell–cell contacts, proliferation, differentiation, and many other fundamental cellular processes [1,2]. Not surprisingly, this family of molecules plays central roles in maintenance of health and their dysregulation often results in disease [3]. In the last few years, important and often surprising insight into the in vivo functions of Rho GTPases has been gained. An essential function of Rho GTPases, only recently elucidated, is the regulation of processes important for the maintenance of whole body metabolic homeostasis, in particularly glucose metabolism and blood glucose control via their actions in metabolically active tissues, such as skeletal muscle and adipose tissue, as well as the pancreas.

Today metabolic diseases significantly contribute to early death in Western society. More than 422 million people worldwide are estimated to have diabetes causing 3% of global deaths [4]. Metabolic diseases also contribute to 1/3 of all cancers [5] and cancer-related deaths [6]. Type 2 diabetes is a metabolic disease that is associated with obesity, reduced insulin-stimulated glucose uptake by skeletal muscle and adipose tissue, and impaired β cell function [7]. Because of their important roles in those processes, Rho GTPases represent a hitherto understudied potential for novel ways to understand dysfunctions in metabolic disease. This review will discuss the key evidence for the role of Rho GTPases in metabolic control and delineate the underlying molecular mechanisms. Further, this is extrapolated to discuss the role of Rho GTPases in conditions of metabolic dysfunction, such as type 2 diabetes and ageing.

### 1.1. Mechanisms of Action

#### 1.1.1. General Mechanisms of Activation

Wide interest in small GTPases was triggered by the discovery of the Ras oncogenes in 1982 [8] and with its mutations a few years later proven to cause human cancers [9,10]. Those discoveries were soon followed by the identification of related proteins now forming the Ras superfamily [11,12]. Rho GTPases function as molecular switches that cycle between an inactive guanosine diphosphate (GDP)-bound and an active guanosine triphosphate (GTP)-bound state (Figure 1). At least three major types of regulatory proteins/factors are described for Rho GTPases as illustrated in Figure 1. Activation of Rho GTPases is induced by guanine nucleotide exchange factors (GEFs), which facilitate switching from a GDP- to a GTP-bound state. Inactivation results from GTPase-activating proteins (GAPs) that promote the intrinsic GTPase activity mediating GTP to GDP hydrolysis [13,14]. Additionally, Rho guanine dissociation inhibitors (RhoGDIs) bind to the inactive, GDP-bound form of Rho GTPases maintaining their inactive state [15,16]. Thus, in order for a biological event to activate Rho GTPases, the balance between the GEFs, GAPs, and RhoGDIs must be altered to promote the Rho GTPase GTP-bound state. Mammalian cells encode genes for numerous of GEFs (>80) and GAPs (>60), but only three RhoGDIs [17]. Depending on the biological mode of regulation, release from RhoGDI must occur and GEFs must be activated to interact with the Rho GTPase and promote GTP binding. Conversely, inactivation occurs when GAPs promote the GTP hydrolysis. Rho GTPases can then be extracted from the membrane by RhoGDIs.

In addition to the regulation of the GTP–GDP cycling, post-translational modifications (e.g., isoprenylation, palmitoylation, phosphorylation, ubiquitination, sumoylation) of Rho GTPases appear to be additionally required for optimal regulation of Rho GTPase function [18,19]. The intracellular spatiotemporal distribution of Rho GTPases is tightly controlled. In general, active Rho GTPases are targeted to the plasma membrane (or endosomal membranes) via a polybasic region and a prenyl group attached to a C-terminal cysteine residue, and several Rho GTPases are also palmitoylated [18,20,21]. Prenylated cytosolic Rho GTPases are unstable and rapidly degraded [22]. Additionally, active Rho GTPases can be targeted for proteasomal degradation by ubiquitination [19,23]. The molecular chaperone, RhoGDI, binds to prenylated Rho GTPases forming a cytosolic pool of mainly inactive GDP-loaded Rho GTPases and protects Rho GTPases from ubiquitination and degradation [22]. Thus, while release from RhoGDI is a necessary event for the Rho GTPases to be activated, the release may reduce Rho GTPase protein abundance. Like Rho GTPases, GEFs, GAPs, and RhoGDIs are also regulated by post-translational modifications, revealing the many modalities for regulation of these key molecular switches [19]. Yet another layer of complexity is added to the regulation of the spatiotemporal distribution and activation, as Rho GTPases can interact to regulate each other’s activity. For example, the Rho GTPases, Rac1/Cdc42, and RhoA have opposing roles and double-negative feedback loops exist between them [24,25,26]. The mechanisms underlying this feedback regulation between Rho GTPases are not resolved but might involve Rho GTPase competitive binding to RhoGDI [22]. One study found that in vascular smooth muscle cells, phosphorylation of RhoA on S188 induced dissociation of Rac1 from RhoGDIα, thereby activating Rac1 and vascular smooth muscle cell migration [27], while protecting RhoA from ubiquitin/proteasome-mediated degradation [28]. Moreover, feedback loops exist between downstream effector proteins and the Rho GTPases. The downstream effectors of Rac1/Cdc42, p21-activated kinases (PAKs), have been reported to negatively regulate GEFs [29,30], while PAK1 phosphorylation of RhoGDI at site S101 and S174 leads to RhoGDI dissociation from Rac1 [31]. Additionally, pharmacological inhibition (IPA-3) of PAK1-3 lowered Rac1 activity [24]. This also increased RhoA activity in agreement with the above mentioned negative regulation between the Rho GTPases.

As outlined above, the regulation of Rho GTPases is highly complex making it challenging to determine the exact regulatory mechanisms operating under any one circumstance. 

#### 1.1.2. Downstream Effects

Active Rho GTPases recruit downstream effector molecules to the plasma membrane and trigger their activation [32]. Conserved structure and mechanism in multiple versions of Rho GTPases in bacteria, yeast, flies, and vertebrates, suggest that all derive from a single primordial protein, repeatedly modified in the course of evolution to perform a variety of functions. Each Rho GTPase affects numerous downstream proteins, all of which have specific roles in various cell processes [32,33]. Especially Rac1, Cdc42, and RhoA control pathways central to metabolic regulation. Rho GTPases are crucial organizers of the actin cytoskeleton (Figure 2). Active GTP-bound RhoA, Rac1, and Cdc42 promote the assembly of actin stress fibers and focal adhesions, thin sheet-like lamellipodial protrusions and finger-like filopodial protrusions, respectively [34,35,36,37,38,39]. The regulation of the actin cytoskeleton by Rho GTPases are extensively reviewed in [1,40,41] and only selected regulatory mechanisms investigated in relation to metabolic regulation will be presented here.

Actin remodeling is controlled through parallel actin polymerization and depolymerization. Generally, Rac1 and Cdc42 regulate comparable downstream events via activation of PAKs and activation of the Wiskott–Aldrich syndrome family of proteins including Wiskott-Aldrich Syndrome protein (WASP), neural-WASP (N-WASP), and WASP family verprolin homologous protein 1 or 2 (WAVE1/2). WASP and N-WASP are critical downstream effectors of Cdc42 that mediate the formation of filopodia through activation of the actin polymerization factor Arp2/3 [42]. Likewise, Rac1 exerts its effect on the cytoskeleton by binding to the WAVE complex, releasing WAVE and thereby promoting lamellipodia polymerization via activation of the Arp2/3 complex [43].

Among other downstream effectors, PAKs activate LIM domain kinases (LIMKs) promoting phosphorylation and inactivation of cofilin [42]. Active cofilin severs and depolymerizes actin filaments and leads to an increase in uncapped barbed ends promoting actin polymerization [42,44]. To counteract LIMK-dependent inactivation of cofilin, Rac1 activates the cofilin phosphatase, slingshot1 (SSH1). Thereby, phosphorylation of cofilin is tightly regulated by Rac1 by two opposite mechanisms, which results in cofilin dephosphorylation and activation [45,46,47]. Rac1 is also an essential component for the assembly of the NADPH oxidase complex (NOX2) by binding to p67-phox, and thereby facilitates the production of reactive oxygen species (ROS) at cell membranes [48,49].

Critical downstream effectors of RhoA are Rho-associated protein kinase (ROCK) and the Formin family of proteins that promotes the formation of straight, unbranched actin fibers. ROCK phosphorylates a large cohort of actin-binding proteins and intermediate filament proteins to modulate their functions. For example, ROCK phosphorylates LIMK leading to the phosphorylation and inactivation of the actin severing protein cofilin [50,51]. Formins capable of actin nucleation promote the elongation of pre-existing filaments by removing barbed end capping proteins [52]. Diaphanous formin (mDIa) is the most extensively studied and suggested to regulate microtubule stabilization in addition to its actin polymerization activity [53,54].

Whereas both downstream pathways of Rac1 and Cdc42 seem to be involved in metabolic control, RhoA may via downstream effectors be involved in muscle mass regulation and thereby play an important role in conditions of muscle mass loss. This will be reviewed below with emphasis on knowledge gained from in vivo studies and humans.

## 2. Rho GTPases in Regulation of Glucose Homeostasis

Rho GTPases have important functions during cell development and differentiation (reviewed in [1,55,56,57]). They also play crucial roles in fully differentiated tissues, including the pancreas, skeletal muscle, and adipose tissue (Figure 3). Following a meal, nutrients are absorbed in the intestines and enter the circulation. This results in a postprandial transient increase in blood glucose and free fatty acids. Blood glucose quickly returns to baseline because of an increased secretion of the hormone insulin from the pancreatic β cells. When in circulation, insulin stimulates glucose uptake into skeletal muscle and the adipose tissue together with inhibition of hepatic glucose production. Thereby insulin returns blood glucose back to baseline [58]. Those processes are significantly dysregulated in metabolically dysfunctional conditions, such as type 2 diabetes and obesity [59,60,61]. Both the pancreatic release of insulin and resulting insulin-stimulated uptake of glucose into skeletal muscle and adipose tissue are tightly regulated by complex processes that involve Rho GTPases. Rho GTPases likely also play insulin-independent roles in the maintenance of glucose homeostasis, including exercise-stimulated glucose uptake and beneficial adaptations to exercise training. In the following sections, we will delineate and discuss the current evidence for a role of Rho GTPases with a focus on their roles in maintaining glucose homeostasis.

### 2.1. Insulin Secretion by Pancreatic β Cells

The secretion of insulin is an essential everyday event that occurs following a meal to promote glucose disposition by the muscle and curb hepatic glucose output. Several lines of evidence place Rho GTPases, in particular Cdc42 and Rac1, as important players in glucose-stimulated insulin secretion via their roles in actin reorganization [62,63,64,65] as depicted in Figure 3. Glucose is a hydrophilic molecule that cannot cross the lipid bilayer membrane. High blood glucose is sensed by pancreatic β cells and the glucose taken up through GLUcose transporters (GLUTs; predominantly GLUT1 and GLUT3 in human islets, GLUT2 in rodents). In general terms, this rapidly translates into elevated intracellular ATP, which in turn closes the ATP-sensitive potassium ion channel, leading to depolarization of the cell membrane. Depolarization causes the opening of voltage-gated calcium channels, and insulin vesicles fuse with the cell membrane to release insulin (reviewed in [66]). During the fasting state when levels of blood glucose are low, the actin cytoskeleton restricts the access of secretory insulin-containing vesicles to their release sites. In response to a postprandial increase of blood glucose, the β cell’s actin cytoskeleton is reorganized, enabling insulin-containing vesicle translocation to the plasma membrane. Under normal circumstances, Cdc42 and Rac1 activation is needed for this process [58,63,67]. The precise mechanisms by which high glucose activates Rho GTPases are not clear but in vitro studies have shown that in response to glucose stimulation, Cdc42 is activated, which leads to Rac1 activation via PAK1 [68]. In agreement, isolated islets from pancreatic β cell-specific Rac1 knockout mice showed decreased glucose-stimulated insulin secretion in response to high glucose concentrations [65]. On the other hand, in vitro studies show that expression of constitutively active Cdc42 interferes with β cell differentiation leading to hyperglycemia [63]. Thus, tight regulation of the Rho GTPases must be in place. In vivo work from Thurmond’s group shows that the downstream effector of Cdc42 and Rac1, PAK1, is involved in insulin secretion, as whole body PAK1 knockout mice were glucose intolerant due to reduced insulin production [69].

Overall, it is well established that Rho GTPases play essential roles in normal β cell function and insulin secretion, primarily via the essential role of Cdc42 and Rac1 in insulin-vesicle translocation in response to increased glucose concentration in the circulation. However, the up- and downstream mechanisms remain to be clearly defined.

### 2.2. Rho GTPases are Implicated in β Cell Dysfunction in Metabolic Disease

Type 2 diabetes is associated with defects in insulin secretion whereby pancreatic β cells fail to compensate for peripheral insulin resistance (described below) so that hyperglycemia ensues [70]. Before the onset of type 2 diabetes, insulin resistance causes hyperproduction of insulin to overcome the insulin resistance of muscle, adipose tissue, and the liver. Accelerated insulin production is only tolerated for a limited period of time, after which the insulin-producing β cells dedifferentiate and insulin production is impaired or even stopped [71]. The stage of insulin hyperproduction has been associated with conditions of glucotoxicity and accelerated production of reactive oxygen species (ROS) in the islets. Rac1 is an important regulator of the assembly of the NADPH oxidase (NOX2) complex at the plasma membrane [49]. While ROS production is important for glucose-stimulated insulin secretion, uncontrolled Rac1-mediated superoxide hyperproduction is associated with β cell failure (reviewed in [72]). For example, hyperactivation of Rac1 has been demonstrable in human islets exposed to glucotoxic conditions and in islets derived from the Zucker diabetic fatty (ZDF) rat [73], a model of type 2 diabetes. Indeed, suppression of Rac1 activation protects β cells against noxious effects of glucolipotoxicity and cytokines [73,74]. Thus, Rac1 exerts damaging roles under pathological conditions by inducing NOX2 activity to create excessive oxidative stress, mitochondrial damage, and cell death [72,75]. In support of that, the expression and activation of Rac1 in pancreatic tissue from ob/ob mice, a model for obesity, is increased [76]. Interestingly, islets from donors with type 2 diabetes display an average PAK1 protein loss of 80% compared with non-diabetics [69] and recent work has shown that islets from donors with type 2 diabetes had profound defects in glucose-stimulated Cdc42 and PAK1 activation together with impaired glucose-stimulated insulin secretion [77]. Long-term hyperglycemia causing glucotoxicity has also been linked to RhoA-dependent stress fiber formation and diminished glucose-stimulated insulin secretion [78]. 

Thus, while the Rho GTPases Cdc42 and Rac1 are critical for insulin secretion, pathological hyperactivation of Rac1 and RhoA negatively affects β cell function during times of stress and may thus be targets for preventing β cell failure in diabetes.

### 2.3. Rho GTPases Regulate Glucose Uptake

The Rho GTPases Rac1, RhoA, and Cdc42 are highly expressed in skeletal muscle and adipose tissue and emerging evidence suggests that they have key functions in the regulation of glucose uptake and the maintenance of whole body glucose homeostasis.

#### 2.3.1. Rho GTPases in Insulin Action in Skeletal Muscle

In response to insulin, the majority of dietary glucose is taken up by skeletal muscle [79,80]. Comprising 40–50% of the body, skeletal muscle is essential for maintaining whole body glucose homeostasis. Glucose uptake into skeletal muscle is an actively regulated process that necessitates mobilization and insertion of GLUT4 into the plasma membrane. This mechanism is primarily promoted by insulin and muscle contraction [81,82]. Perhaps not surprisingly due to their major role in vesicle traffic in other cell types, Rho GTPases are involved in the traffic of GLUT4-containing vesicles to the muscle plasma membrane (Figure 3).

Rac1, TC10, and RhoA are activated by insulin in skeletal muscle [83,84,85,86,87]. The first evidence that Rho GTPases are actively involved in the regulation of glucose uptake in skeletal muscle came nearly 20 years ago when the lack of functional Rac1 was observed to attenuate insulin-stimulated GLUT4 translocation in L6 muscle cells in vitro [88,89]. Additionally, constitutive active Rac1, but not Cdc42 or TC10, increased the amount of GLUT4 in the plasma membrane [83,90]. Subsequent studies also demonstrated an important role for Rac1 in insulin action in mature skeletal muscle in vivo. In mouse skeletal muscle, pharmacological inhibition and genetic ablation of Rac1 abolished insulin-stimulated GLUT4 translocation and reduced insulin-stimulated glucose uptake [86,87,91,92].

The involvement of Rac1 in insulin-stimulated GLUT4 translocation has been ascribed to a Rac1-mediated insulin-stimulated actin remodeling in skeletal muscle cells [83,88,89]. However, the relevance of a mobile actin cytoskeleton in fully mature skeletal muscle was recently questioned as mice that lack β-actin, one of the major actin cytoskeleton isoforms in skeletal muscle, displayed normal glucose transport [93]. On the other hand, in mature skeletal muscle, the actin depolymerizing agent, Latrunculin B reduced insulin-stimulated glucose uptake [87,91], suggesting participation of the actin cytoskeleton.

It is largely unknown how Rho GTPases are activated by insulin but it seems to occur downstream of phosphatidylinositol 3-kinase (PI3K). Skeletal muscle evidently expresses the following GEFs for Rac1: Tiam1 [94], Trio [95], Def6 [96], son of sevenless [97], Arf6 [98], Kalirin [99], Vav2, FLJ00068, DBl-I, α- and β-PIX (PAK-interacting exchange factor) [100], and switch-associated protein 70 (SWAP70) [90]. Among these, FLJ00068 has been suggested to regulate insulin-stimulated Rac1-dependent GLUT4 translocation in L6 myotubes [90] and mouse skeletal muscle [101]. Further research within this area will add to the understanding of the mechanism by which Rac1 regulates GLUT4 translocation.

RhoGDIα and RhoGDIβ are expressed in skeletal muscle [102,103] and RhoGDIα (but not RhoGDIβ) forms complexes with Rac1, RhoA, and Cdc42 [15]. Recent data from our laboratory has identified RhoGDIα as a negative regulator of Rac1 activity and GLUT4 translocation in skeletal muscle. In L6-GLUT4myc myotubes, siRNA-mediated RhoGDIα knockdown augments Rac1 activity and GLUT4 translocation both in the basal state and in response to insulin stimulation. Corroborating those in vitro results, RhoGDIα overexpression in mouse skeletal muscle in vivo decreased insulin-stimulated glucose uptake and impaired whole-body glucose tolerance (unpublished work, [104]). Thus, RhoGDIα is a novel key regulator of Rac1, important for maintaining the appropriate level of glucose uptake in muscle. The mechanisms by which insulin triggers Rac1 dissociation from RhoGDIα in skeletal muscle are currently unknown but it could involve phosphorylation of RhoGDIα by PAK or Src as described for other cell types [105,106].

Downstream of Rac1, PAK1 and PAK2 (PAK3 is not expressed in skeletal muscle) are activated in response to insulin [69,87,107] and are suggested to mediate Rac1-dependent insulin-stimulated reorganization of the actin cytoskeleton and GLUT4 translocation in skeletal muscle cells [108]. However, the role of group I PAKs in mature mouse skeletal muscle is debated. One study reported markedly impaired insulin-stimulated GLUT4 translocation in vivo in skeletal muscle lacking PAK1 [69], while recent work in our laboratory found no impairment in insulin-stimulated glucose uptake in PAK1 knockout mouse skeletal muscle in vivo or in isolated muscles in vitro [109]. In transgenic mice jointly lacking PAK1 and PAK2 only a mild reduction in insulin-stimulated glucose transport in muscle was evident [109], suggesting that group I PAKs play only a minor role in the regulation of glucose uptake. In support of this, constitutively activated Cdc42, an activator of group I PAKs, did not stimulate GLUT4 translocation in muscle cells in vitro [90]. The discrepancy between the PAK studies could perhaps be explained by the age of the mice, as double PAK1/2 knockout mice exhibit other age-related phenotypes [110]. Further downstream of Rac1, signaling through Arp2/3 and cofilin could mediate the cortical actin polymerization evoked by insulin (see Figure 2). In L6 myoblasts siRNA-mediated silencing of Arp3 abrogates actin remodeling and impairs GLUT4 translocation [47]. Moreover, cofilin knockdown causes overwhelming actin polymerization that subsequently inhibits GLUT4 translocation [47]. Those findings suggest that most distal regulators of actin cytoskeleton dynamics evoked by Rac1 are involved in insulin-stimulated glucose uptake, although those findings remain to be confirmed in differentiated muscle and in vivo. Thus, while the important role for Rac1 in insulin-stimulated glucose uptake is clear, Rac1 likely mediates glucose uptake via PAK-dependent, but also PAK-independent mechanisms that remain to be elucidated.

An input by RhoA in the regulation of insulin-stimulated glucose transport, has also been proposed, although the results are few and rather inconsistent [111]. Furthermore, all results are obtained by manipulating pathways downstream of RhoA and in several cases using pharmacological inhibitors reported to have unspecific effects [112]. Among the various effectors of RhoA, the kinase ROCK stands out. Two isoforms, ROCK1 and ROCK2, are expressed in skeletal muscle. ROCK1-deficient mice are insulin resistant [113] and siRNA-mediated silencing of ROCK1 reduced insulin-stimulated glucose transport in L6 myotubes [114], suggesting that ROCK1 is a positive regulator of insulin signaling. In addition, overexpression of a dominant negative ROCK2 or pharmacological inhibition of ROCKs impair insulin-stimulated GLUT4 translocation and glucose uptake in muscle in vivo [85].

Challenging a positive input of RhoA in the regulation of glucose uptake, membrane-bound (active) RhoA is elevated in skeletal muscles of obese Zucker rats [115]. In that model, jointly overexpressing a dominant-negative ROCK1 and 2 rescued diet-induced obesity and improved glucose tolerance [116], which contrasts with the findings in mice expressing only the dominant negative ROCK2 [85] or lacking ROCK1 [113] as described above. Interestingly, knockout of geranylgeranyl diphosphate synthase 1 (GGPPS; a branch point enzyme in the mevalonic acid pathway) specifically in skeletal muscle of mice decreased membrane-associated and prenylated RhoA (= active) and improved whole body glucose tolerance [117]. However, GGPPS might also affect the activity of other GTPases regulated by prenylations, including Rac1, which was not investigated. Thus, direct evidence for RhoA’s role in muscle insulin action is still lacking.

Collectively, there is ample support that Rac1, and possibly RhoA, but seemingly not Cdc42, participate in the insulin-dependent regulation of glucose uptake in skeletal muscle.

#### 2.3.2. Cross-Talk Between Rho GTPases and Canonical Insulin Signaling

Canonical insulin signaling via Akt and TBC1D4 is a well-described signaling mechanism by which insulin increases muscle and adipose tissue glucose uptake [118,119]. In muscle and muscle cells in culture, both Akt/TBC1D4 and Rac1 are important regulators of insulin-stimulated glucose uptake. Studies from our laboratories have suggested that Rac1 and Akt mediate insulin-stimulated glucose uptake via parallel signaling mechanisms. This is because, in our hands, silencing Rac1 expression via siRNA [89] or knocking out the Rac1 gene [87] did not affect insulin-stimulated Akt phosphorylation although it abolished GLUT4 translocation [83,86,90]. Moreover, by inhibiting Rac1 or Akt2 separately, each partially reduced insulin-stimulated glucose transport, while joint inhibition of both pathways nearly fully blocked it [91]. Thus, the Rac1 and Akt pathways seem to act on GLUT4 translocation through distinct downstream processes. On the other hand, studies from the Satoh group point to cross-talk or feedback loops between the Rac1 and the Akt signaling pathways. For example, basal (but not insulin-stimulated) Akt activity was required for constitutively active Rac1 to induce GLUT4 translocation in L6 myotubes [90]. In agreement, Rac1 and PAK have been reported to be involved in the activation of Akt in various cell types [120,121] suggesting the possible existence of a feedback loop between Rac1 and Akt. Also, several publications from Satoh´s group suggest that Akt2 is involved in insulin-stimulated Rac1 activation leading to glucose uptake in muscle [101,122,123]. Most recently, this was proposed because knockdown of Akt2 by siRNA abolished Rac1 activation following intravenous administration of insulin or ectopic expression of a constitutively activated PI3K mutant in mouse muscle [124]. Similarly, we have observed reduced insulin-stimulated phosphorylation of PAKs in extensor digitorum longus, but not soleus muscle from Akt2 KO mice, although this was not observed in any of the muscles in response to pharmacological Akt inhibition using MK2206 [91]. Thus, currently the relative role of each pathway and how they interact to regulate insulin-stimulated glucose uptake is not fully understood. In our hands, a superactivation of Rac1 achieved by an acute overstimulation of the Rho GTPase lead to Akt phosphorylation, but the levels of Rac1 superactivation are far higher than those physiologically stimulated by insulin [125]. Based on these collective results we propose that, in muscle, the level of Rac1 activation achieved with insulin does not suffice to stimulate Akt2, however, it is possible to show a connection between these signals when Rac1 activation is supraphysiological. Moreover, while Akt might have some regulatory role over Rac1, Rac1 activity does not rely on Akt. Rac1 can be activated independent of Akt, since exercise and muscle contraction, which do not activate Akt, activate Rac1.

#### 2.3.3. Exercise Elicits Metabolic Benefits via Regulation of Rho GTPases

It has been convincingly documented that regular exercise improves glycemic control and insulin action among healthy, as well as obese and type 2 diabetic subjects [126,127,128,129], and that this effect can be superior to those exerted by drugs or insulin therapy [130,131]. Physical activity causes a large increase in energy utilization [126,132,133]. Glucose is a major fuel source for the contracting muscles and glucose uptake acutely increases more than 50-fold during exercise [133]. Because glucose is taken up by the contracting muscles via insulin-independent mechanisms [134,135], it is effective in lowering blood glucose in insulin resistant subjects [136].

Rac1 is activated during exercise in mouse and human skeletal muscle [137] and Rac1 muscle-specific knockout mice exhibit reduced exercise-stimulated glucose uptake in muscle [138,139]. Likewise, GLUT4 translocation in response to electrical pulse stimulation (which mimics, but does not recapitulate muscle contraction) is reduced by inhibition of Rac1 in C2C12 myotubes [140].

Very recent results shed light onto the downstream events by which Rac1 may induce glucose uptake during exercise. Henriquez-Olguin et al. showed that Rac1 is implicated in the increase in ROS-production that occurs during exercise in muscle [141]. As described for pancreatic β cells, Rac1 is an integral part of the NOX2 complex in skeletal muscle. In skeletal muscle Rac1-mediated production of ROS contributes to increased glucose uptake in mouse skeletal muscle during exercise [141]. Because mice lacking PAK1 displayed normal contraction-stimulated glucose transport in skeletal muscle [109], one can assume that Rac1 does not mediate contraction-stimulated glucose uptake via PAK1. Rac1 would then be available for activation, contributing to NOX2 activation. The resulting ROS production would in turn contribute to the stimulation of muscle glucose uptake [141] by an unknown mechanism. Thus, Rac1 helps remove glucose from the blood via insulin independent mechanisms.

In addition to the acute stimulation of glucose uptake by a single bout of exercise, exercise training also induces remarkable skeletal muscle adaptations that benefit metabolic regulation, including the enhancement of insulin sensitivity (recently reviewed in [126]). Recent work has implicated RhoA in exercise-training adaptations, as seen by the increased expression of ROCK1, ROCK2, and RhoA in rat gastrocnemius muscle following short-term swimming exercise training [142]. Furthermore, ROCK1/2 inhibition by Y-27632 impaired the insulin sensitizing effect of exercise training [142]. Although non-specific effects of pharmacological inhibitors cannot be ruled out, that study suggests that the downstream effectors of RhoA and ROCK1/2 are involved in the metabolic benefits of exercise training.

Interestingly, exercise results in phosphorylation of RhoGDIα on serine 34 in human skeletal muscle [143]. Whether RhoGDIα serine 34 phosphorylation in muscle destabilizes its interaction with Rho GTPases to permit their activation during exercise will be interesting to determine.

In summary, exercise elicits metabolic benefits, several of which involve regulation of Rho GTPases. Future studies should delineate the mechanisms by which Rho GTPases may be influencing health with exercise and importantly if they can be explored pharmaceutically to harness some of the metabolic benefits of exercise in individuals that cannot adhere to an exercise program.

### 2.4. A Possible Role for Rho GTPases in Adipose Tissue Glucose Uptake

Although being mainly a fat storage tissue, adipose tissue is also essential for maintaining metabolic homeostasis and health. Especially in conditions of obesity, adipose tissue takes up a substantial amount of glucose following a meal, thereby significantly contributing to the clearance of glucose from the blood. Like in skeletal muscle, adipocyte glucose uptake relies on the translocation of GLUT4 to the plasma membrane for glucose to diffuse across the membrane lipid bilayer. Insulin activates TC10 [144,145], Cdc42 [146], Rac1 [147], and RhoA [148] in adipocytes but their involvement in adipose tissue glucose uptake is not clear.

An early investigation showed that transfection of a constitutively-active RhoA increased GLUT4 translocation in response to insulin, while dominant-negative RhoA significantly decreased it in 3T3-L1 and rat adipocytes [149]. Another study using siRNA-mediated Cdc42 knockdown reported that Cdc42 could mediate insulin signaling to glucose transport in 3T3-L1 adipocytes [146]. 3T3-L1 adipocytes seemingly respond normally to insulin after transfection of a dominant-negative or constitutively-active Rac1 [150]. However, Marcusohn et al. did not confirm that the mutants, transfected in the fibroblast stage of the culture, persisted in the adipocyte stage where insulin-stimulated glucose uptake was studied. Nevertheless, that study was supported by the finding that pharmacological Rac1 inhibition did not reduce insulin-stimulated glucose uptake in incubated fat pads from mice [87], although actual inhibition of Rac1-signaling was not confirmed in that study. Thus, despite methodological issues, the few studies to date suggest no role for Rac1 in insulin stimulated glucose uptake in adipocytes.

The Rho GTPase TC10 also seems relevant for regulating insulin-stimulated glucose uptake in adipose tissue as depicted in Figure 3. Dominant-negative TC10 prevents actin reorganization [151,152], an event that is necessary for adipocytes to translocate GLUT4 to the membrane. Accordingly, dominant-negative TC10 inhibits adipocyte glucose uptake and GLUT4 translocation [144]. That study [144] also reported no effect on insulin-stimulated GLUT4 translocation and glucose uptake of transfecting constitutively-active, or dominant-negative RhoA or Cdc42 in adipocytes (although data was not shown), contrasting with the other studies described above [146,149].

Taken together, Rho GTPases could play roles in adipocyte glucose uptake by regulating GLUT4 vesicle dynamics. Indeed, the fusion of GLUT4 vesicles with the adipocyte membrane requires dynamic insulin-induced actin polymerization, evincing the contribution of the actin cytoskeleton at different steps in the process of GLUT4 translocation [153]. However, the studies linking Rho GTPases to this event are few and the results conflicting with only TC10 consistently shown to regulate insulin-stimulated GLUT4 translocation in adipocytes. Importantly, mechanistic in vivo evidence is completely lacking and there is currently no human evidence in the literature. Thus, the involvement of Rho GTPases in adipose tissue function is largely unexplored and exciting discoveries lie ahead.

### 2.5. Rho GTPases are Implicated in Insulin Resistance—A Key Contributor to Metabolic Disease

In insulin resistant human subjects that are obese or have type 2 diabetes, insulin-stimulated glucose uptake by skeletal muscle is a primary site of dysfunction [79]. Due to their key role in skeletal muscle (and perhaps adipocyte) glucose uptake, Rho GTPases are relevant in metabolic diseases, including type 2 diabetes. Decreased skeletal muscle Rac1 signaling has been associated with insulin resistance in insulin resistant obese and type 2 diabetic human subjects [87], as well as in diabetic ob/ob mice and diet-induced obese insulin resistant rodents [91]. This has also been proposed by in vitro studies of muscle showing that ceramide-induced insulin resistance was associated with marked impairments in insulin-induced Rac1 activation [89]. However, in another in vitro model of insulin resistance, palmitate treatment did not impair Rac1 activity, although the phosphorylation of Rac1’s downstream target, PAK1 was reduced [154]. Interestingly, in a recent study from our laboratory, we observed that lack of Rac1 in muscles from diet-induced obese mice exacerbates insulin resistance [155], suggesting a relevance for Rac1 in counteracting metabolic dysfunctions. Those studies highlight the possibility that compromised Rac1 activity and/or downstream signaling contribute to the development of muscular insulin resistance.

RhoA might also regulate glucose homeostasis, as pharmacological inhibition of RhoA’s target, ROCK, using Fasudil prevented high-fat diet-induced hypercholesterolemia and glucose intolerance in mice [156]. Furthermore, body weight, serum lipid levels, and glucose metabolism were improved in mice with whole body overexpression of a dominant-negative ROCK, compared with littermate control mice [116]. In muscle cells in vitro, palmitic acid-induced insulin resistance was associated with increased expression of ROCK1 [157]. However, that ROCK would negatively affect glucose homeostasis contradicts the fact that lean ROCK1-deficient mice are insulin resistant [113] and lean mice with overexpression of a dominant negative ROCK2 display impaired insulin-stimulated GLUT4 translocation and glucose uptake in muscle in vivo [85]. Indeed, the RhoA-ROCK pathway seems to be clinically relevant, since insulin-stimulated muscle ROCK1/2 activity was attenuated in obese and type 2 diabetic subjects during an euglycemic hyperinsulinemic clamp compared to lean control [158]. Thus, the results on RhoA’s role in metabolic (dys)regulation warrant clarification.

Taken together, Rho GTPases, in particular Rac1 and RhoA, may be highly implicated in muscular insulin resistance and thereby contribute to metabolic dysregulation in metabolic diseases, such as type 2 diabetes.

## 3. Rho GTPases as Hitherto Unrecognized Regulators of Muscle Mass

Muscle mass is important for metabolic health because it increases the amount of tissue available for storing glucose. Although studies are few, there is emerging evidence that Rho GTPases may play hitherto unrecognized roles in muscle mass regulation. Indeed, Rho GTPases are highly regulated during conditions of muscle atrophy. For example, a marked reduction in levels of RhoA was noted in the muscles of unweighted hindlimbs (an experimental model simulating muscle atrophy under weightlessness) in mice [159] and in dystrophic mice [160] along with rapid atrophy. In male Sprague-Dawley rats, three days of hindlimb suspension-induced muscle atrophy decreased RhoA mRNA and protein expression [161]. In contrast, muscle mass loss caused by denervation, increased RhoA expression but returned to baseline when the decline in muscle mass ceased [162]. Thus, RhoA expression is highly regulated in models of muscle mass loss although this remains to be recapitulated in human skeletal muscle [163]. Conversely, increasing muscle mass using functional overload and anabolic steroid administration in mice [164], or hypertrophy-stimulating resistance training in humans elevated the expression of skeletal muscle RhoA [165]. Future studies should investigate muscle mass regulation in muscles that lack RhoA in order to directly determine the implications for RhoA.

Rho GTPases other than RhoA may also be involved in muscle mass regulation as proteomic analysis of mouse muscles following denervation-induced atrophy identified significant changes in RhoGDIα, PAK1-2, and Cdc42 expression [166]. Dominant-negative Cdc42, introduced with a retroviral vector, resulted in fibers that appeared atrophic [167] and blocking Rac1 function in precursors of the indirect flight muscle of *Drosophila* severely disrupted muscle formation. Thus, Rac1 is involved in the regulation of myoblast proliferation and segregation during adult myogenesis [168]. Downstream of Rac1 and Cdc42, PAK1 and PAK2 are activated during mammalian myoblast differentiation. Combined deletion of PAK1 and PAK2 results in reduced muscle mass, a phenotype that is exacerbated after repair to acute injury [110,169]. In support of that, pharmacological inhibition of group I PAKs (with IPA-3) delays skeletal muscle regeneration following cardiotoxin injury in vivo [170], suggesting that Rho GTPase-mediated signaling is important for muscle regeneration. Interestingly, insulin-stimulated phosphorylation of PAK1 at threonine 423 [171] and PAK1 protein content [171,172] were markedly increased in follistatin-induced hypertrophic mouse muscle compared to controls.

A role for Rho GTPases in muscle mass regulation is perhaps not surprising given their well-known requirement for tumor growth. Looking ahead, lessons from the tumor literature may help to understand the mechanisms by which Rho GTPases may be involved in muscle mass regulation in connection with metabolic regulation.

### The Role of Rho GTPases in Muscle Wasting Conditions

Skeletal muscle atrophy is a severe consequence of ageing and many chronic diseases, including cancers. Muscle strength is inversely related to death from all causes [173] and is of the utmost importance for the preservation of mobility and quality of life. RhoA and RhoGDIα are both upregulated in mouse skeletal muscle with age-related muscle mass loss [174]. In agreement, single muscle fiber proteomics analysis showed that RhoGDIα protein expression increased with age in both slow and fast muscle fibers from human biopsy samples, while RhoA increased with age predominantly in fast muscle fibers [175]. Importantly, age-related muscle atrophy only occurred in the fast muscle fiber types. However, contradicting those two studies, a recent study found reduced RhoA protein expression in skeletal muscle of middle-aged rats together with diminished levels of ROCK proteins [176]. Thus, Rho GTPases might be differentially regulated at different ages and stages of sarcopenia, and this warrants further investigation.

Many cancers are associated with cachexia, a condition of involuntary body weight loss including severe muscle atrophy that is not due to anorexia [177]. Interestingly, PAK1 mRNA and protein expression are reduced in cancer-associated cachectic muscles from colon adenocarcinoma C26-bearing mice [170], although PAKs’ upstream Rho GTPases, Rac1, and Cdc42 were not examined. That is consistent with the role of group I PAKs in muscle mass regulation [110,169]. Indeed, PAK1 overexpression partly preserved fiber size in cachectic muscles [170], suggesting that the defect in PAK might be directly involved in the pathogenesis. From these collective studies, RhoA, Rac1 and Cdc42, and PAK emerge as candidate regulators of muscle mass. However, studies exploring a direct mechanistic role for the Rho GTPases in muscle mass regulation are completely lacking.

As muscle is the largest organ of the body, completely necessary for mobility and also responsible for the majority of glucose disposal, future studies should investigate the role for Rho GTPases in muscle wasting diseases. 

Unresolved issues are showed in Box 1.

Box 1Unresolved issues.Unresolved Issues
Lack of in vivo experiments to support the in vitro literature on Rho GTPase regulation and in particular their role in metabolism.Evidence on the regulatory functions of Rho GTPases in humans is limited.Molecularly, the upstream activators and downstream effectors of Rho GTPases in different tissues and in response to different stimuli are poorly defined.Cross-talk between Rho GTPases is poorly defined but important to delineate, as they challenge all interpretation of data using knockdown or overexpression of a single Rho GTPase.High throughput methodological advances to directly measure in vivo GTP binding (fast hydrolysis) warranted.Whether Rho GTPases can be targeted pharmacologically to benefit metabolic diseases remains to be determined.


## 4. Conclusions

In this review, we summarize evidence for the role of Rho GTPases in metabolic regulation in health and disease. We demonstrate that Rho GTPases may be hitherto overlooked players in glucose homeostasis by contributing to metabolically essential functions in skeletal muscle, adipose tissue, and the pancreas. However, this area of research is at its early stages and mechanistic in vivo insights are lacking. This will be an exciting area for future discoveries.

## Figures and Tables

**Figure 1 cells-08-00434-f001:**
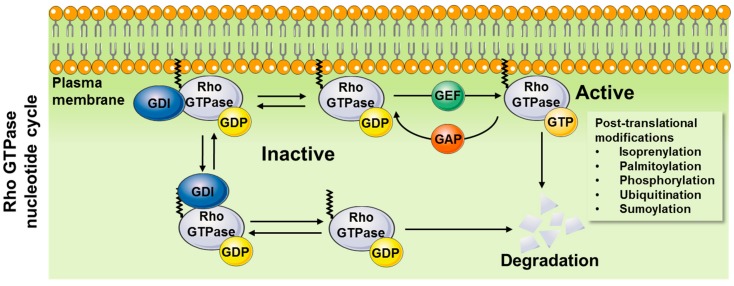
Rho guanosine triphosphatase (GTPase) nucleotide cycle. Activation of Rho GTPases require: i) release from Rho guanine dissociation inhibitors (GDIs) and ii) activation of guanine nucleotide exchange factors (GEFs), which facilitate switching from an inactive guanosine diphosphate (GDP)-bound to an active guanosine triphosphate (GTP)-bound state and translocation to the plasma membrane. Inactivation occurs when GTPase-activating proteins (GAPs) stimulate GTP to GDP hydrolysis and the Rho GTPase re-binds to RhoGDI. Free prenylated Rho GTPases are unstable and degraded and also active Rho GTPases can be targeted to degradation.

**Figure 2 cells-08-00434-f002:**
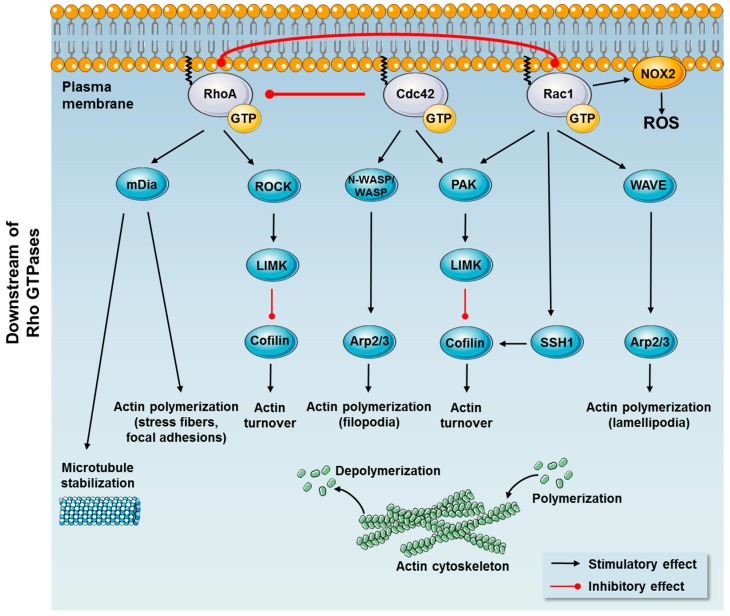
Downstream effector proteins of the major Rho GTPases, Rac1, Cdc42, and RhoA. Downstream effector proteins of Rac1 and Cdc42 include the Wiskott–Aldrich syndrome family of proteins (Wiskott-Aldrich Syndrome protein (WASP), neural-WASP (N-WASP), WASP family verprolin homologous protein 1 or 2 (WAVE1/2)) and p21-activated kinases (PAKs). WASP, N-WASP, and WAVE promote actin polymerization via activation of the Arp2/3 complex. PAKs activate LIM domain kinases (LIMKs) promoting phosphorylation and inactivation of the actin severing protein cofilin. Rac1-dependent activation of slingshot1 (SSH1) promotes cofilin dephosphorylation and actin depolymerization (actin turnover). Downstream effector proteins of Rac1 also include the NADPH oxidase complex (NOX2) to produce reactive oxygen species (ROS) at the cellular membrane. Downstream effector proteins of RhoA include Rho-associated protein kinase (ROCK) and the formin family of proteins (mDia). Among other downstream effector proteins, ROCK regulates the actin turnover via the LIMK–Cofilin pathway.

**Figure 3 cells-08-00434-f003:**
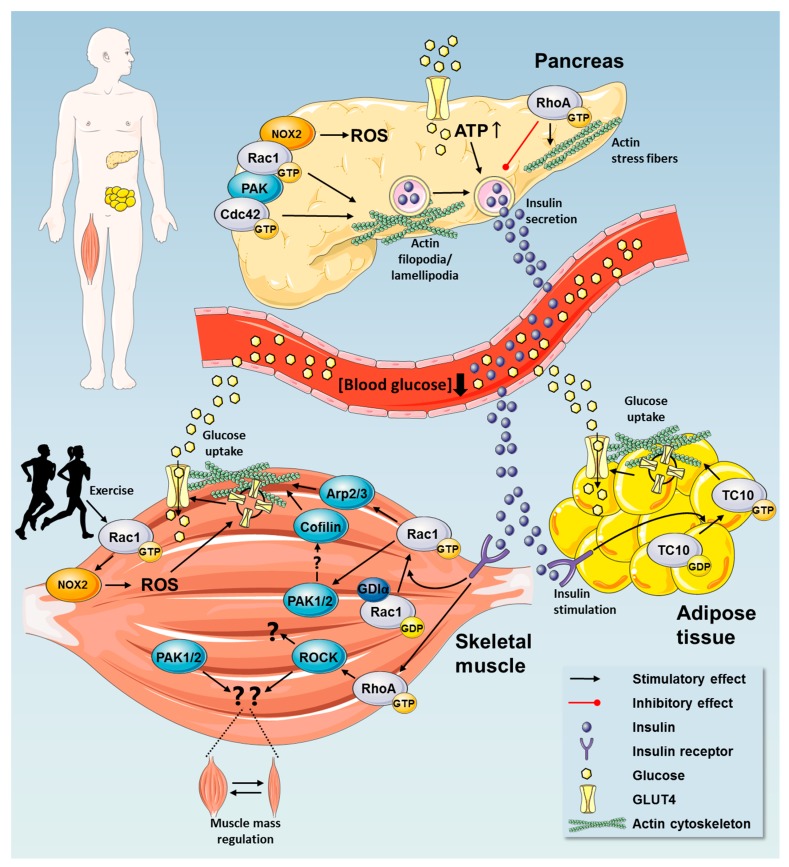
Rho GTPases play tissue-specific crucial roles in fully differentiated tissues, including the pancreas, skeletal muscle, and adipose tissue with the main physiological outcome of lowering blood glucose.

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
