# Peer review of "Rho GTPases—Emerging Regulators of Glucose Homeostasis and Metabolic Health"

_cells, 2019, doi:10.3390/cells8050434_

Round 1

Reviewer 1 Report

This is a comprehensive and timely review on the role of Rho GTPases in glucose homeostasis and metabolic health that is guaranteed to be of interest to the readers of “Cells.” The authors have performed an exhausting review of the literature and have performed an excellent review of the current knowledge in this topic.

The following minor changes would improve the review:

1.     GEFs definition is “Guanine nucleotide exchange factors” and not “Guanine exchange factors.”

2.     Line 61: “Mammalian cells encode genes for numerous of GEFs and GAP” should give a ball park number for the number of genes encoding Rho GTPase GEFs, i.e. >90.

3.     Line 97: It is not entirely true that “strategies to investigate the regulation of Rho GTPases in vivo are still wanting.” Several  Studies have used dominant active and dominant negative expression, small molecule inhibitors ( reviewed in Cancer Res. 2018 Jun 15;78(12):3101-3111; see  Diabetologia. 2015 Nov;58(11):2573-81. Used Vav/Rac inhibitor EHop-016 to study glucose stimulated insulin secretion in the pancreas) and fluorescence resonance energy transfer (FRET) (Cell Adh Migr. 2014;8(6):526-34.) to study Rho GTPase effects.

4.     Line 223:” Thus, while the Rho GTPases Cdc42 and Rac1 is critical for insulin secretion” change to “Thus, while the Rho GTPases Cdc42 and Rac1 are critical for insulin secretion.”

5.     Line 267-268: The assertion that the “The mechanisms by which insulin triggers Rac1 dissociation from RhoG_D_I_α remain to be explored.” needs to be qualified, others have shown this to be due to phosphorylation of RhoGDI by Src and PAK kinases activated by the insulin receptor: see Mol Biol Cell. 2006 Nov;17(11):4760-8; Mol Cell. 2004 Jul 2;15(1):117-27;  J Biol Chem. 2010 Feb 26;285(9):6186-97. 

6.     Line 337: “to the stimulation of muscle glucose uptake by unknown mechanism.” Change to “to the stimulation of muscle glucose uptake by an unknown mechanism.”

7.     Line 401: “pharmacological inhibition of RhoAs target, ROCK using Fasudil prevented” change to “pharmacological inhibition of RhoA’s target, ROCK, using Fasudil prevented”

8.     Line 408-409: “However, that ROCK would negatively affect glucose homeostasis contradict the fact” change to “However, that ROCK would negatively affect glucose homeostasis contradicts the fact”

9.     Line 421: “in the muscles of unweighted hindlimbs in mice” did the authors mean to say “under weight?”

10.  Line 451: “is of outmost  importance” change to “is of utmost  importance”

11.  Line 464: “Although PAKs” change to “Although PAK’s”

Author Response

Response to reviewer 1.

This is a comprehensive and timely review on the role of Rho GTPases in glucose homeostasis and metabolic health that is guaranteed to be of interest to the readers of “Cells.” The authors have performed an exhausting review of the literature and have performed an excellent review of the current knowledge in this topic.

We thank the reviewer for evaluating our review and for constructive feedback. We have addressed all of the points raised as indicated below. Thank you for taking the time to go over our manuscript.

The following minor changes would improve the review:

1.     GEFs definition is “Guanine nucleotide exchange factors” and not “Guanine exchange factors.” Thank you for spotting this, it has now been corrected.

2.     Line 61: “Mammalian cells encode genes for numerous of GEFs and GAP” should give a ball park number for the number of genes encoding Rho GTPase GEFs, i.e. >90. In the mentioned sentence, we have now included the approximate number of human Rho GTPase GEFs and GAPs.

3.     Line 97: It is not entirely true that “strategies to investigate the regulation of Rho GTPases in vivo are still wanting.” Several  Studies have used dominant active and dominant negative expression, small molecule inhibitors ( reviewed in Cancer Res. 2018 Jun 15;78(12):3101-3111; see  Diabetologia. 2015 Nov;58(11):2573-81. Used Vav/Rac inhibitor EHop-016 to study glucose stimulated insulin secretion in the pancreas) and fluorescence resonance energy transfer (FRET) (Cell Adh Migr. 2014;8(6):526-34.) to study Rho GTPase effects. We agree that inhibitors are available and we meant to state that it is difficult to directly assess GTPase activity in vivo. However, we state that point later inbox 1 and we have deleted this sentence here as it might be a bit misplaced.

4.     Line 223:” Thus, while the Rho GTPases Cdc42 and Rac1 is critical for insulin secretion” change to “Thus, while the Rho GTPases Cdc42 and Rac1 are critical for insulin secretion.” This has been corrected.

5.     Line 267-268: The assertion that the “The mechanisms by which insulin triggers Rac1 dissociation from RhoG_D_I_α remain to be explored.” needs to be qualified, others have shown this to be due to phosphorylation of RhoGDI by Src and PAK kinases activated by the insulin receptor: see Mol Biol Cell. 2006 Nov;17(11):4760-8; Mol Cell. 2004 Jul 2;15(1):117-27;  J Biol Chem. 2010 Feb 26;285(9):6186-97. We agree that this is a possibility. However, to the best of our knowledge this never been explored in skeletal muscle in response to insulin, which is what we intended to point out. The studies suggested were performed in other cell types. However, we have now included a short description of such studies to open up for that possibility.

6.     Line 337: “to the stimulation of muscle glucose uptake by unknown mechanism.” Change to “to the stimulation of muscle glucose uptake by an unknown mechanism.” Thank you, this is now corrected

7.     Line 401: “pharmacological inhibition of RhoAs target, ROCK using Fasudil prevented” change to “pharmacological inhibition of RhoA’s target, ROCK, using Fasudil prevented” This has been corrected

8.     Line 408-409: “However, that ROCK would negatively affect glucose homeostasis contradict the fact” change to “However, that ROCK would negatively affect glucose homeostasis contradicts the fact” This has been corrected, thank you for spotting this.

9.     Line 421: “in the muscles of unweighted hindlimbs in mice” did the authors mean to say “under weight?” We actually mean “unweighted”. It means that the leg does not carry any weight and therefore undergoes atrophy. This comment has prompted us to rephrase to make that clearer.

10.  Line 451: “is of outmost importance” change to “is of utmost  importance” This has been changed

11.  Line 464: “Although PAKs” change to “Although PAK’s” Thank you for spotting this. It has been corrected to “Although PAKs’ upstream Rho GTPases…”. According to the Oxford Dictionaries Online (https://en.oxforddictionaries.com/punctuation/apostrophe), a plural noun that already ends in –s, should have an apostrophe added after the s.

Reviewer 2 Report

General comments:

In this manuscript the authors present a thorough review of the still limited but significant data on the emerging role of Rho GTPases in regulating several aspects of whole body metabolic homeostasis, namely glucose metabolism and blood glucose control. Importantly, they also discuss how the dysregulated activity of these ubiquitous signaling regulators can contribute to the etiology of metabolic diseases such as diabetes, which represent a significant public health problem in western society.

Particular comments:

The manuscript is well written, with an adequate structure and a fluid style that makes for a easy reading.

Aside from a few typing errors, which are listed below, my only criticism to the manuscript is that topic 2.2 would benefit from an additional subtopic discussing the crosstalk between Rho GTPase signalling and the AKT/AS160/Rab GTPases axis, in the overall regulation of GLUT4 plasma membrane abundance and glucose uptake.

Minor issues:

Page 1:

-          Check font size in lines 41 and 42;

Page 2:

-          Line  89  â€“ “effector proteins” instead of “effecter proteins”;

-          Line 90 – “p21-activated kinases (PAKs), has have been reported”;

Page 3:

-          Line 111 – remove spaces in “Wiskott - Aldrich”. Also throughout the manuscript;

Page 7:

-          Line188 – space missing in “Rac1activation”;

-          Line 189 – standardize the β symbol used throughout the manuscript;

-          Line 193 - Pak1 was already defined so here "the downstream effector" fits better than "a downstream effector"...;

Page 8:

-          Line 234 – GLUT4 was defined above;

Page 9:

-          Line 302 –  The authors refer to a dominant-negative of which ROCK isoform?

-          Line 309 – “RhoA’s” instead of “RhoAs”;

Page 11:

-          Line 401 – “RhoA’s” instead of “RhoAs”;

Page 12:

-          Lines 462 to 465: “Consistent with the role of group I PAKs 462 in muscle mass regulation [95,146], PAK1 mRNA and protein expression are reduced in muscle in 463 cancer-associated cachexia in colon adenocarcinoma C26-bearing mice [147], although PAKs 464 upstream Rho GTPases, Rac1 or Cdc42 was not examined.” Confusing sentence!

Author Response

Response to Reviewer #2

In this manuscript the authors present a thorough review of the still limited but significant data on the emerging role of Rho GTPases in regulating several aspects of whole body metabolic homeostasis, namely glucose metabolism and blood glucose control. Importantly, they also discuss how the dysregulated activity of these ubiquitous signaling regulators can contribute to the etiology of metabolic diseases such as diabetes, which represent a significant public health problem in western society.

We thank the reviewer for evaluating our review and for constructive feedback. We have addressed all of the points raised as indicated below. Thank you for taking the time to go over our manuscript.

Particular comments:

The manuscript is well written, with an adequate structure and a fluid style that makes for a easy reading.

Aside from a few typing errors, which are listed below, my only criticism to the manuscript is that topic 2.2 would benefit from an additional subtopic discussing the crosstalk between Rho GTPase signalling and the AKT/AS160/Rab GTPases axis, in the overall regulation of GLUT4 plasma membrane abundance and glucose uptake. We thank the reviewer for constructive criticism, input and suggestions. We have addressed them all as indicated below and have included a new subtopic in 2.2 on the possible crosstalk between RhoGTPase signaling and the Akt signaling pathway. There are some controversies in the literature, mainly between ours and another group’s findings, however, we have attempted to present the literature in a balanced fashion.

Minor issues:

Page 1:

-          Check font size in lines 41 and 42; Thank you, this has been corrected

Page 2:

-          Line  89  â€“ “effector proteins” instead of “effecter proteins”; Thank you, this has been corrected

-          Line 90 – “p21-activated kinases (PAKs), has have been reported”; Thank you, this has been corrected

Page 3:

-          Line 111 – remove spaces in “Wiskott - Aldrich”. Also throughout the manuscript; Thank you, this has been corrected

Page 7:

-          Line188 – space missing in “Rac1activation”; This has been corrected

-          Line 189 – standardize the β symbol used throughout the manuscript; This has been corrected

-          Line 193 - Pak1 was already defined so here "the downstream effector" fits better than "a downstream effector"...; We agree, this has been corrected

Page 8:

-          Line 234 – GLUT4 was defined above; Thank you, this has been corrected

Page 9:

-          Line 302 –  The authors refer to a dominant-negative of which ROCK isoform? Tracking back the generation of the mice, we found that it was actually both isoforms. This has been clarified in the text. Thank you for pointing out this important matter.

-          Line 309 – “RhoA’s” instead of “RhoAs”; This has been corrected

Page 11:

-          Line 401 – “RhoA’s” instead of “RhoAs”; This has been corrected

Page 12:

-          Lines 462 to 465: “Consistent with the role of group I PAKs 462 in muscle mass regulation [95,146], PAK1 mRNA and protein expression are reduced in muscle in 463 cancer-associated cachexia in colon adenocarcinoma C26-bearing mice [147], although PAKs 464 upstream Rho GTPases, Rac1 or Cdc42 was not examined.” Confusing sentence! Thank you for pointing this out. We have attempted to clarify the sentence.